# HF Radars for Wave Energy Resource Assessment Offshore NW Spain

Ana Basañez *[ID] and Vicente Pérez-Muñuzuri [ID]

Group of Nonlinear Physics, CRETUS Institute, University of Santiago de Compostela, 15782 Santiago de Compostela, Spain; vicente.perez.munuzuri@usc.es
* Correspondence: anabmercader@gmail.com

**Abstract:** Wave energy resource assessment is crucial for the development of the marine renewable industry. High-frequency radars (HF radars) have been demonstrated to be a useful wave measuring tool. Therefore, in this work, we evaluated the accuracy of two CODAR Seasonde HF radars for describing the wave energy resource of two offshore areas in the west Galician coast, Spain (Vilán and Silleiro capes). The resulting wave characterization was used to estimate the electricity production of two wave energy converters. Results were validated against wave data from two buoys and two numerical models (SIMAR, (Marine Simulation) and WaveWatch III). The statistical validation revealed that the radar of Silleiro cape significantly overestimates the wave power, mainly due to a large overestimation of the wave energy period. The effect of the radars' data loss during low wave energy periods on the mean wave energy is partially compensated with the overestimation of wave height and energy period. The theoretical electrical energy production of the wave energy converters was also affected by these differences. Energy period estimation was found to be highly conditioned to the unimodal interpretation of the wave spectrum, and it is expected that new releases of the radar software will be able to characterize different sea states independently.

**Keywords:** HF radar; wave energy; wave modeling; remote sensing; wave energy converter; resource characterization

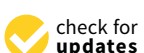



## 1. Introduction

The development of renewable energies is one of the key factors in the fight against greenhouse gas emissions, reduction in waste, and for promoting a diverse and distributed energy mix. The least widespread, but with great potential, are the marine renewable energies, among which offshore wind energy is the most developed technology, followed by tidal and wave energy [1]. Wave energy offers the greatest potential for exploitation, mainly due to its spatial availability and low needs of investment in infrastructure [1,2], but it is also the most complicated to assess [3]. Theoretically, the global power potential of waves is approximately 1000–10,000 GW, covering the global power demand which is ≈1800 GW/year (or 16,000 TWh/year energy consumption) [4,5]. The northwest coast of Galicia (NW Spain) is considered one of the areas of the Iberian Peninsula with the largest wave energy resource, e.g., offshore Costa da Morte wave energy is estimated to be approximately 400 MWh/m per year [6], and slightly further South, offshore Cape Silleiro, it is approximately 300 MWh/m [7].

Wave energy is harnessed by wave energy converters (WECs), which currently are mostly engineering projects and prototypes [4,8,9], with a particular electricity production rate depending on wave quantity and qualitative characteristics (height, period, and direction) [10]. Therefore, a gross estimate of the available wave energy may not represent the production of a WEC [11–14].

Hence, the wave energy resource assessment must imply an accurate description of the wave regime characteristics in the areas of interest to select the most optimal location

for the wave farms, and moreover, for managing the energy to be supplied to the power grid [9,15].

The uncertainties in this assessment have two key elements: (i) on the one hand, the method to estimate the wave spectral parameters (height, period and direction) can consider that all sea states follow a unimodal wave spectrum, or, on the contrary, to consider this as bi- or multimodal when simultaneous wave systems show up (e.g., swell or wind sea) and to be decomposed into different spectra to represent each wave system [10,16,17]. Furthermore, the temporal scope must be chosen according to the temporal variability of the wave resource in the area of study (annual, seasonal, monthly) [7,18–20]. (ii) The second is how to collect the most accurate wave data. Buoys are the traditional method of wave measurement and are considered the most reliable in situ wave gauges. However, they have certain drawbacks such as installation and maintenance. Moreover, buoys are single-point measuring devices and since they occupy a space that competes with other uses, it is not possible to anchor them massively. Therefore, the current most widespread method to obtain wave data with wide coverage and spatial resolution are numerical models. Among them, WAM (Wave modeling) and WW3 (WaveWatch III) have been widely developed for deep water wave assessment [5,15,21–24] and SWAN (Simulating waves nearshore) for calculating wave propagation towards the coast, where bathymetry and shoreline characteristics are more relevant and higher resolution is required [7,15,25–27].

Although these models are under continuous development and adjustment [28–30], their outcomes are not direct measurements. Moreover, there is some complexity involved in their application such as considering multiple factors and particularities of the area of interest [31] and the hindcast data characteristics, such as the sampling frequency which determines the accuracy of the temporal variability described by the model [22,32,33]. Likewise, the outcomes of SWAN depend on the wave patterns selected for its propagation [3,29]. As an alternative to models and buoys, new remote sensing technologies such as the oceanographic high-frequency radars (HF radar) are currently being applied to obtain current and wave data with high spatial resolution [34–39]. Radar technology is based on the Doppler effect that the wave emitted by the radar experiences as it backscatters from the ocean surface [40]. The radar software generates a spectrum with the backscatter signal as a function of its frequency shift. This describes first- and second-order peaks, from which the radial component of the currents and the directional spectra of the waves are extracted [41,42].

Despite many works having shown the efficiency of these radars [36,43–47] some limitations have also been described [39,48,49]. Firstly, the signal only returns to the radar when it backscatters from waves (called Bragg waves) with a wavelength one-half that of the radar and it propagates towards or away from the radar. Additionally, the second-order peaks only rise at the Doppler spectra when the Bragg waves are modulated by waves with greater periods [44]. The radar working frequency (or the wavelength) also determines the minimum and maximum measurable wave height [50,51], as well as their sensitivity to the current's speed that could prevent the correct estimation of the wave parameters [50]. Secondly, although new radar software to separate the spectrum is under development [52], in general, the adjustment of the wave spectrum is based on unimodal models of the sea state [53–55]. Moreover, in the case of the Seasonde model radar, it is also assumed that the measured area is in deep waters and the swell is uniform throughout [50]. Moreover, according to many works, the centroid period estimated by these radars usually takes values between the buoys-derived mean and peak period [56,57]. Finally, the radar data can be modulated by tides [56], as it happens to the buoys [58].

The two radars which are the focus of this work are the Seasonde models of CODAR. These have been proven to produce reliable wave data, especially regarding spectral significant wave height [36,39,59,60], but also some disadvantages, such as the lack of detection of most of the waves below 3 m [39], a significant disagreement with buoy data regarding wave period and some deviations of the wave direction [36,39,59,60]. However, some of these issues might be determined by the uncertainties that each location can bring

to the validations, such as the heterogeneity of the sea state and the complexity of the coverage area [36,39].

The use of HF radars for wave energy resource assessment has been scarce. With an OSCR radar, the wave height was used to describe the wave energy in Miami [61]. With a WERA radar, the wave power was obtained in the WaveHub area (Cornwall, UK) with a spatial resolution of ≈600 m [55,62]. Moreover, with a WERA radar, in the central coast of Chile, the wave power was evaluated using the significant wave height provided by the radar but using the wave period as a constant value based on previous works [63]. With a Seasonde radar, the inter and intra-annual variability of the energy resource in Galway Bay (Ireland) was analyzed using only the radar-derived wave height [19].

HF radars are wave-measuring devices with wide spatial and temporal coverage. Moreover, their installation on land allows easy access for data collection and instruments' maintenance. Therefore, we consider that HF radars could be a useful tool for wave energy resource assessment, as well as provide great support for numerical models' validation. In this work, we made a preliminary analysis of using two HF radars wave data for wave energy resource assessment, at the offshore areas of Vilán and Silleiro capes (NW of Galicia, Spain). To evaluate the results, most of the analyses were replicated using the wave data from two nearby buoys and two numerical models, namely SIMAR (Simulación Marina—Marine Simulation in Spanish) and WW3 (Figure 1). The wave power and energy calculated with the radars' data were statistically validated against the buoys and SIMAR model. The wave energy resource was described through energy matrices, the annual and seasonal mean energy and power roses. The key differences between the radars and the other sources of data results were detailed. Likewise, the spatial distribution of the mean wave energy calculated with the radars' data was compared with the WW3 model. The previous wave energy resource description was used to calculate the theoretical electrical energy production of two WECs, and again the radar estimates were compared with the ones from the buoys and SIMAR model. Finally, the viability of using data from Vilán cape radar to complete the discontinuous Vilano-Sisargas buoy data series was analyzed.

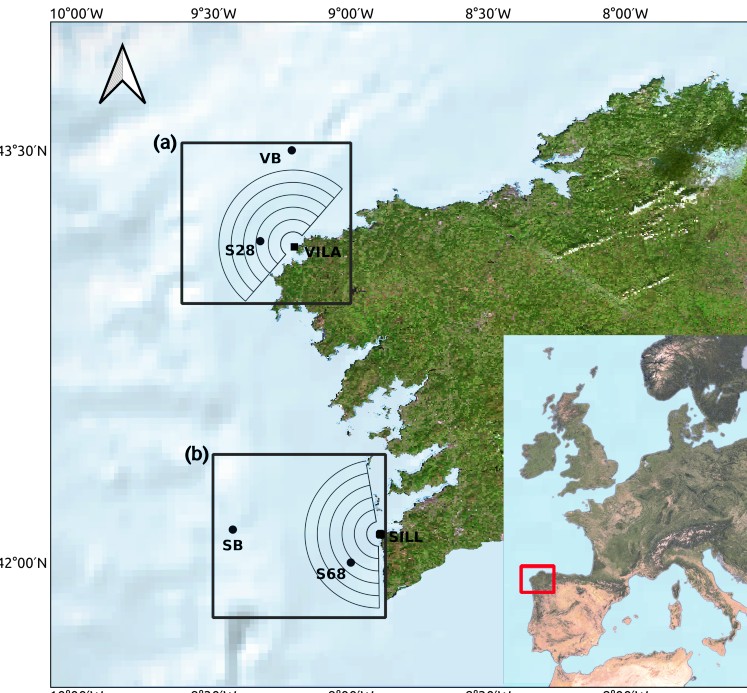

**Figure 1.** Northwestern coast of Galicia (NW Spain) with Vilán (**a**) and Silleiro (**b**) areas of study. The following data sources are indicated in the figure: **VB**: Vilano-Sisargas buoy; **VILA**: Vilán HF radar and range cells of width of 5 km (RCs); **S28**: SIMAR (Marine Simulation) point 3004028; **SB**: Silleiro buoy; **SILL**: Silleiro HF radar and range cells; and **S68**: SIMAR point 1044068.

## 2. Materials and Methods

The HF radars used for this study were two CODAR Seasonde radars: the cape Vilán radar (VILA) managed by Intecmar-Xunta de Galicia and the cape Silleiro radar (SILL) managed by Puertos del Estado (PdE) (Figure 1). These are long-range wide beam radars (emitting in all directions), whose working frequency is 4.86 MHz with a sweep of ≈29.41 kHz. Each one has two antennas, one to emit and the other to receive the backscatter produced on the ocean surface. The latter antenna is made up of three antennas arranged at 90° angles between each other so that the difference in the signal received by each of them will be used in the process of identifying the direction of origin of the signal. The accuracy of the 'direction-finding' method [53] ranges between 2° and 30° [64,65].

The scanning area was between 5 km and 30 km away from the radar. Since the spatial resolution of these radars is 5 km, [66] the signal was processed independently for five concentric rings, 5 km wide, around the radar called range cells (RCs) (Figure 1). Considering the radars emit and receive in all directions, the CODAR's proprietary software configuration sets the so-called coastal limits (CL) according to the characteristics of the radar location, so that only the signal coming from the open sea is processed. In the case of VILA, these are 221° and 41°, and for SILL, these are 180° and 350° (see Figure 1). The same limits are applied to waves direction (wave bearing limits, WB) [50,67].

The set of signals received during 180 min was processed to form the averaged Doppler spectra [64]. Then, the wave frequency spectrum was calculated and fitted to the Pierson–Moskowitz model to extract the spectral significant wave height ($H_{m0}$) and centroid period ($T_c$). The mean wave direction ($D_m$) was calculated by applying a cardioid direction factor to the equation of the wave spectrum [53]. Although the initial signal processing time was 180 min, these parameters were calculated at every 30 min of the swell.

Considering the previously discussed limitations of these radars [44] and the CODAR recommendations [64], the software was configured to process waves with periods between 5 and 17 s. When the received signal was too low to calculate the wave parameters, the radar software flags the corresponding samples [64], which we will refer to hereafter as nulls. Filtering based on the quality control defined by the Copernicus Marine in situ TAC (Thematic Assembly Centre) [39,68] was applied to the radar data series. In addition, considering Basañez et al. [39], we flagged 2020 results as fails, corresponding to the samples that met the following conditions. Although they were not very abundant, they represented very high $H_{m0}$ and $D_m$ errors:

- VILA: $D_m$ North–Northwest (NNE) (330°–90°) and $H_{m0} \geq 6$ m;
- SILL: $D_m$ South (S) (180°–235°) and $H_{m0} \geq 5$ m.

To validate the radar data, wave data from two buoys managed by PdE were used: the Vilano-Sisargas buoy (VB), located at approximately 40 km Northwest of Cape Vilán; and the Silleiro buoy (SB), approximately 60 km West of Cape Silleiro (Figure 1). Both buoys are SeaWatch models that record their movements over the water surface (relative accelerations and directions) for 26 min. The data were processed in situ, calculating the net displacements of the buoys and a wave spectrum was generated. As a result, every hour a sample was generated providing, among others, the wave spectral parameters [69]. These were received in real-time by PdE and subjected to the Copernicus Marine In Situ TAC quality control method [68], which assigns to each sample a code based on its reliability. From the resulting files, $H_{m0}$, the peak period ($T_p$) and $D_m$, as well as the energy period ($T_e$), calculated from the raw spectral buoy data, were used.

The SIMAR points, provided by PdE, were part of the WANA subset (waves Analysis), developed by PdE and the Agencia Estatal de Meteorología (AEMET). WANA produces wave data using the WAM (Wave Modelling) and WaveWatch models, and these, in turn, use the HIRLAM model (High Resolution Local Area Modelling) as wind forcing. It is important to emphasize that SIMAR data are not forecasts but a reanalysis provided by the models considering all locations as open and deep waters [70]. For this work, we used wave data from SIMAR points 3004028 (S28) and 1044068 (S68), whose positions were

within the RC between 10 and 15 km (RC 10 km) for both radars (Figure 1). The spectral parameters were $H_{m0}$, $T_p$ and $D_m$.

The other model was the WW3, whose data were provided by Meteogalicia. This model provides a wide spatial coverage of the two study areas (Figure 1) and a spatial resolution of 0.5° (≈5 km). WW3 uses the GFS (Global Forecast System) as meteorological forcing. The outputs of this model contain hourly wave samples; the first 12 h corresponding to a reanalysis and the following 96 h to the forecast period [22]. The wave spectral parameters ($H_{m0}$ and $T_p$) used in this work correspond to the reanalysis period.

The data series from both models were subjected to a quality control process by the operators (PdE, Meteogalicia), whereby unreliable samples were marked with specific codes.

The data period used for this work spans from 1 January 2014 to 7 October 2020. Data availability is a limitation factor on radars and buoys since the SIMAR data series are only missing a few days and the WW3 data series is considered complete. For the radars, the lack of data was due to problems in the operation or adjustments of their installation, and in the case of the buoys, mainly due to maintenance periods. Although there are abundant gaps during small periods of time (around hours or days), there are also others that span from months to years, as is the case of VILA and SILL data series (Figure 2).

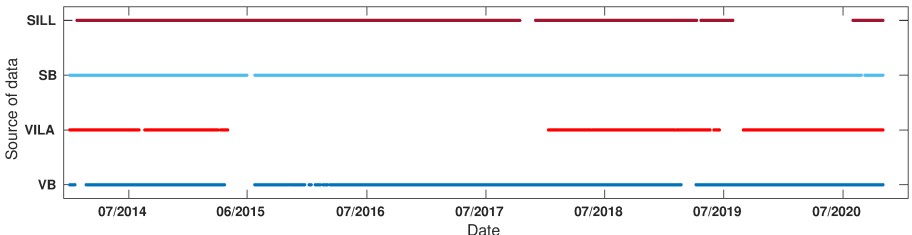

**Figure 2.** Timeline of the available raw samples of Silleiro radar (**SILL**), Silleiro buoy (**SB**), Vilán radar (**VILA**) and Vilán buoy (**VB**). The relevant gaps are the following: **SILL**: 13 October–11 November 2017; 8–28 April 2019; 28 July–8 January 2020. **SB**: 28 June–23 July 2015; 26 August–6 September 2020. **VILA**: 1–19 August 2014; 31 March–9 April 2015; May 2015–January 2018; 18–24 May 2018; 20 May–1 June 2019; 1 June 1–30 August 2019. **VB**: 17 January–20 February 2014; 21 April–22 July 2015; 23 December 2015–9 March 2016; 20 February–7 April 2019; 29 July 2019–1 August 2020.

The radars' data availability was also affected by the large number of samples marked as nulls, which on average, were 60% of the data (Table 1), but could be more as the distance to the radar increases and the wave height diminishes [39]. In Basañez et al. [39], we observed that VILA null data correspond to almost all the waves below 2 m, 50% of which were between 2–3 m and around 20–40% were up to 5 m. In the case of SILL, the largest data loss occurred for waves with $H_{m0}$ below 2 m, but above this value, the percentage of nulls was much lower than for the VILA case.

Considering this interdependence between the wave conditions, the amount and even the quality of the data [39], as well as the difference in the available periods of data for the Vilán and Silleiro areas, we decided to analyze the wave energy resource of each area independently. The study areas are those bounded by the boxes in Figure 1, and the time periods used for each one are:

- Vilán area: 1 January 2014–30 April 2015 + 1 January 2018–7 October 2020.
- Silleiro area: 1 January 2014–28 July 2019 + 1 August 2020–7 October 2020.

Table 1 shows the ideal number of hourly samples that would correspond to each period, as well as the number of raw samples available from each data source for the whole period, and the sum for each season of the year for the whole period (spring: April–June; summer: July–September; autumn: October–December; winter: January–March). For the radars, the number corresponds to 30 min samples and the percentage of nulls is also detailed.

**Table 1.** Total and seasonal hourly raw samples of SIMAR points and buoys (S28; S68; VB; SB); the ideal number of hourly samples for each period; 30 min raw samples of the 10 km RCs for the radars (VILA, SILL) and their percentage of null data (Nulls). WW3 (WaveWatch III model) has the same sample size as ideal samples.

| (a) | VILÁN SITE | | | | |
|---|---|---|---|---|---|
| Data Sets | S28 | VB | Hourly Ideal Samples | VILA 10 km | Nulls |
| Total | 35,666 | 33,637 | 35,879 | 63,839 | 51.05% |
| Spring | 9420 | 9059 | 9456 | 15,742 | 55.72% |
| Summer | 8737 | 8759 | 8832 | 13,858 | 71.40% |
| Autumn | 6734 | 6768 | 6767 | 13,497 | 45.05% |
| Winter | 10,775 | 9051 | 10,824 | 20,742 | 37.81% |
| (b) | SILLEIRO SITE | | | | |
| Data Sets | S68 | SB | Hourly Ideal Samples | SILL 10 km | Nulls |
| Total | 50,377 | 49,596 | 50,471 | 95,202 | 39.38% |
| Spring | 13,092 | 13,053 | 13,104 | 25,280 | 46.01% |
| Summer | 13,188 | 12,377 | 13,199 | 25,891 | 57.01% |
| Autumn | 11,173 | 11,183 | 11,184 | 19,770 | 29.41% |
| Winter | 12,924 | 12,983 | 12,984 | 24,261 | 21.79% |

For deep waters, wave power (W/m) was calculated as [2,3,71]

$$P = \frac{\rho g^2}{64\pi} H_{m0}^2 T_e \tag{1}$$

where $\rho$ is the seawater density and $g$ is the acceleration of gravity. $T_e = m_{-1}/m_0$ is the energy wave period, and $m_{-1}$ and $m_0$ are the spectral moments $-1$ and $0$, respectively. For those cases where the wave spectra of the other data sources were not available, $T_e = 0.8572 T_p$ [11,27,72].

For the validation of the total wave power and energy, paired clean data from radars, buoys and SIMAR points were used. The objective of this method was, in combination with previous results [39], to statistically compare the differences between the variables, without interference from radar nulls or any gaps in the data series. Unfortunately, the data set size drops considerably (Table 2).

**Table 2.** Paired clean samples for each area and period.

| (a) | VILÁN SITE | (b) | SILLEIRO SITE |
|---|---|---|---|
| | Paired Clean Samples | | Paired Clean Samples |
| Data Sets | All Sources | Data Sets | All Sources |
| Total | 13,724 | Total | 27,486 |
| Spring | 3210 | Spring | 4656 |
| Summer | 1849 | Summer | 6618 |
| Autumn | 3650 | Autumn | 6891 |
| Winter | 5015 | Winter | 9321 |

To describe the wave energy resource, energy matrices, annual and seasonal mean energy and power roses were used. Paired raw samples of radars, buoys and SIMAR points were used. Therefore, raw samples were the same for all data sources of each site (29,578 for Vilán site and 46,475 for Silleiro site). These, unlike the previous case, include the radar

nulls and samples marked by quality filters. Since the calculations were made only with the valid samples, the size of each data source used were different as it is shown in Table 3. With this method, in addition to other differences, the aim was to detect the incidence of the data loss, especially from the radars, to calculate the wave energy resource.

**Table 3.** Paired raw data of each area and the total and seasonal valid samples (not nulls or fails) of each data source. The size of WW3 model data set is the same as the SIMAR model.

| (a) | | VILÁN SITE | | (b) | | SILLEIRO SITE | |
|---|---|---|---|---|---|---|---|
| Paired Raw Data: 29,578 Samples. | | | | Paired Raw Data: 46,475 samples. | | | |
| Valid Samples | | | | Valid Samples | | | |
| Data Sets | S28 | VILA RC 10 km | VB | Data Sets | S68 | SILL RC 10 km | SB |
| Total | 29,578 | 13,770 | 29,537 | Total | 46,475 | 27,856 | 45,768 |
| Spring | 7439 | 3224 | 7425 | Spring | 12,501 | 6627 | 12,488 |
| Summer | 6,819 | 1,859 | 6,814 | Summer | 12,076 | 4997 | 11,412 |
| Autumn | 6723 | 3657 | 6,718 | Autumn | 9848 | 6898 | 9835 |
| Winter | 8597 | 5030 | 8580 | Winter | 12,050 | 9334 | 12,033 |

Energy matrices are grids representing the percentage of occurrence ($OP_{ht}$) of the number of hourly ideal samples ($I$) of each combination of $T_e$ vs. $H_{m0}$ (cell resolution $1\,\text{s} \times 1\,\text{m}$). On top of these graphs, five lines were superimposed, representing the power (kW/m) of the cell they crossed and also corresponding to the 25th, 50th, 75th, 90th, and 99th percentiles of each power data set [19]:

$$
\begin{aligned}
OP_{ht} &= O_{ht} \times 100 \\
O_{ht} &= M_{ht}/I
\end{aligned}
\tag{2}
$$

where $M_{ht}$ is the number of samples corresponding to each cell. $I$ is the same for all data sources, which allows to compare between the energy matrices and detect the effect of radar nulls. The mean energy $Em$ (Wh/m) was calculated by multiplying the annual or seasonal hours, corresponding to each cell (hours occurrence, $OH_{ht}$) of a matrix of resolution $0.5\,\text{s} \times 0.5\,\text{m}$, by the mean power ($P_{ht}$) of the samples corresponding to each cell ($P_j$) [3,27]:

$$
\begin{aligned}
Em &= \sum_{ht=1}^{C} P_{ht} \times OH_{ht} \\
P_{ht} &= \left(\sum_{j=1}^{M_{ht}} P_j\right) / M_{ht} \\
OH_{ht} &= O_{ht} \times H
\end{aligned}
\tag{3}
$$

where $C$ is the total number of cells and $H$ is the ideal number of hours of the period for which the average power is to be calculated (8760 h/year and 2190 h/season).

The method used to estimate the electrical output from a WEC device consisted of summing the result of multiplying the WEC power matrix by an hour's occurrence matrix, like those described above [32]. The power matrix represents the electrical power (W) that the WEC can generate for each cell ($H_{m0}$ vs. $T_p$ or $T_e$). These usually have significant restrictions regarding minimum and maximum $H_{m0}$ and $T$ values, considering their technology or their own safety [3]. For this work, power matrices were generated for Pelamis and Aquabuoy engines, based on data previously published [13,73].

## 3. Results

### 3.1. Statistical Validation of Wave Power and Energy

$T_e$, wave power and wave energy were validated with the paired clean data for each data source (see Table 2). A detailed validation of $H_{m0}$ and $T$ samples calculated for the two radars and buoys was previously developed [39], which revealed a significant linear correlation of $H_{m0}$ (larger for VILA case $\approx 0.88$), but also that it is slightly overestimated by the radars, except for the smallest and highest waves which are underestimated compared to the buoy-derived parameters [39]. The $T_p$ linear correlation was smaller than for $H_{m0}$, and in general, the radars overestimate $T_p$ except for the largest values that are clearly underestimated. Hence, in this work, we focused only on carrying out the validation of the $T_e$ data series. Thus, the agreement of VILA $T_e$ data with the VB ones is larger than for $T_p$. However, there is a VILA overestimation trend except for some range of buoy $T_e$ samples which are underestimated by the radar (Figure 3a). For SILL, the underestimation of the extreme values of $T_e$ was lower than that of $T_p$, but still relevant, and the SILL overestimation of the bulk of the samples was very noticeable (Figure 3c). The SIMAR points seem to underestimate the lowest buoy $T_e$ data, but as $T_e$ increases, SIMARs data overestimate $T_e$ (Figures 3b,d).

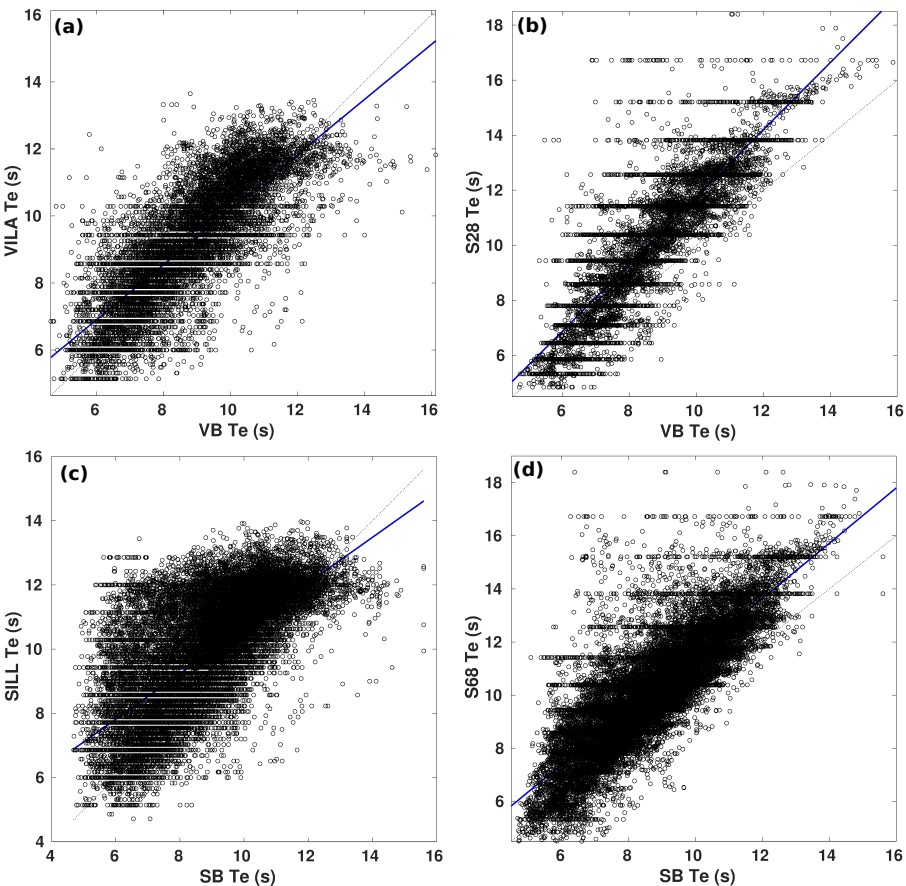

**Figure 3.** $T_e$ scatter plots of Vilano-Sisargas buoy (VB) vs. Vilán radar (VILA) (**a**), and SIMAR S28 (**b**); and Silleiro buoy (SB) vs. Silleiro radar (SILL) (**c**) and SIMAR S68 (**d**). Blue lines correspond to the best data linear fitting.

Total energy values for each data source were calculated (Figures 4–7) and the mean power for each period and data set, as well as the RMSE between them, are shown in Tables 4 and 5.

VILA shows a slight overestimation of the annual wave energy compared to VB data (Figure 4). However, this does not seem to follow a clear trend when seasonal data are compared, as shown in Figure 5. The average seasonal wave power (Table 4) confirms

this overestimation in all cases, which seems to be a percentage-wise larger in spring and summer. Something similar occurs when comparing results from S28 and VB. On the other hand, although the mean power from VILA and S28 is quite similar, RMSEs between both data series are larger than when compared to VB data.

**Table 4.** Mean wave power and wave power root mean square error (RMSE).

| VILÁN SITE | | | | |
|---|---|---|---|---|
| **Wave Power Statistics (kW/m)** | | | | |
| **Data Sets** | | **MEAN** | **RMSE vs. VB** | **RMSE vs. S28** |
| **Total** | **VB** | 67.00 | — | — |
| | **S28** | 69.48 | 29.37 | — |
| | **VILA** | 71.76 | 41.73 | 44.17 |
| **Spring** | **VB** | 32.47 | — | — |
| | **S28** | 37.34 | 15.88 | — |
| | **VILA** | 39.18 | 23.75 | 23.20 |
| **Summer** | **VB** | 22.47 | — | — |
| | **S28** | 22.95 | 8.74 | — |
| | **VILA** | 24.83 | 19.54 | 20.63 |
| **Autumn** | **VB** | 84.06 | — | — |
| | **S28** | 87.92 | 33.18 | — |
| | **VILA** | 93.70 | 47.93 | 52.14 |
| **Winter** | **VB** | 92.91 | — | — |
| | **S28** | 93.79 | 37.00 | — |
| | **VILA** | 93.95 | 49.93 | 52.58 |

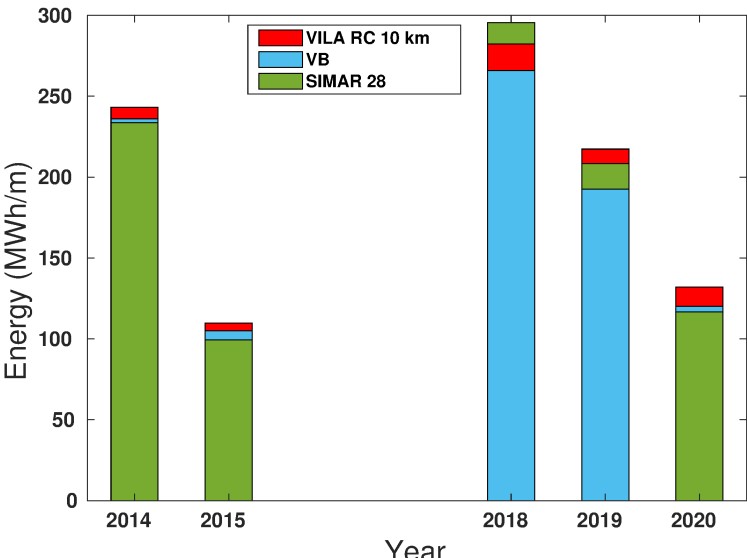

**Figure 4.** Superimposed bars of total wave energy calculated for VILA (**red**); S28 (**green**); and VB (**blue**) per year.

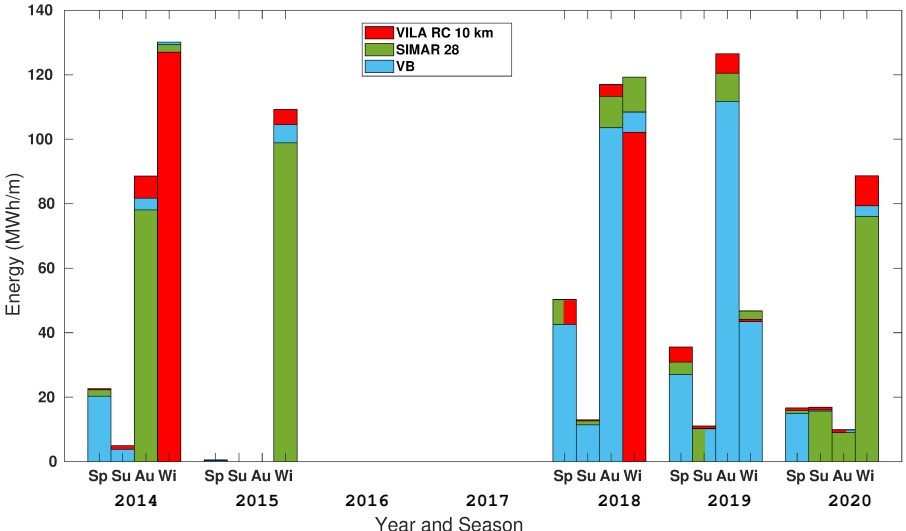

**Figure 5.** Superimposed bars of total wave energy calculated with VILA (**red**); S28 (**green**); and VB (**blue**) per season of the year.

The wave energy calculated with the SILL radar is always larger than that estimated with its corresponding buoy and SIMAR point (Figure 6) and the highest overestimation seems to occur in winter (Figure 7). However, the wave power validation (Table 5) shows that the largest SILL overestimation occurs in spring and summer. Except in spring 2018, the energy estimated with the S68 point data is always lower than that estimated with SILL and SB. This is also confirmed by the mean wave power comparison (Table 5).

**Table 5.** Mean wave power and wave power root mean square error (RMSE).

| | SILLEIRO SITE | | |
|---|---|---|---|
| | **Wave Power Statistics (kW/m)** | | |
| **Data Sets** | **MEAN** | **RMSE vs. SB** | **RMSE vs. S68** |
| **Total** | | | |
| SB | 54.39 | — | — |
| S68 | 45.14 | 28.87 | — |
| SILL | 66.00 | 40.02 | 46.57 |
| **Spring** | | | |
| SB | 29.97 | — | — |
| S68 | 27.33 | 14.59 | — |
| SILL | 40.36 | 27.94 | 30.76 |
| **Summer** | | | |
| SB | 19.22 | — | — |
| S68 | 15.36 | 9.37 | — |
| SILL | 29.40 | 24.69 | 27.54 |
| **Autumn** | | | |
| SB | 60.74 | — | — |
| S68 | 49.44 | 27.92 | — |
| SILL | 76.76 | 44.23 | 53.13 |
| **Winter** | | | |
| SB | 84.61 | — | — |
| S68 | 69.48 | 41.07 | — |
| SILL | 94.51 | 49.17 | 57.07 |

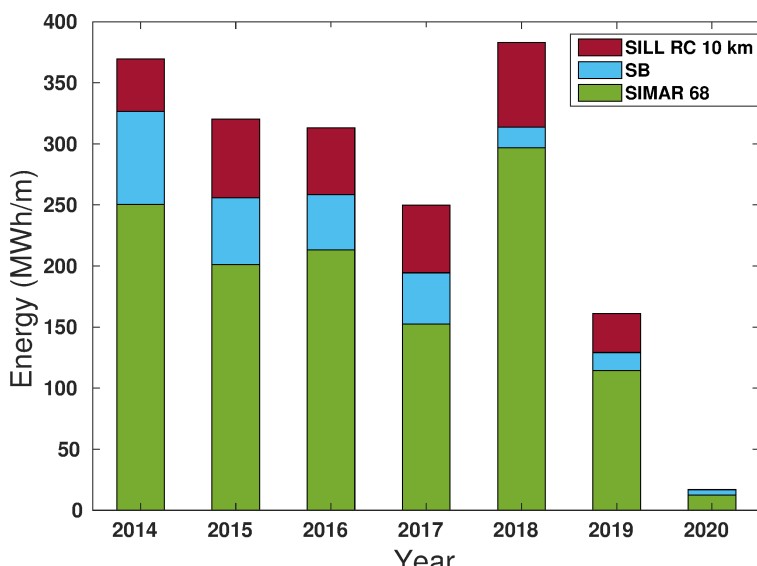

**Figure 6.** Superimposed bars of total wave energy estimated with SILL (**red**), S68 (**green**) and SB (**blue**) per year.

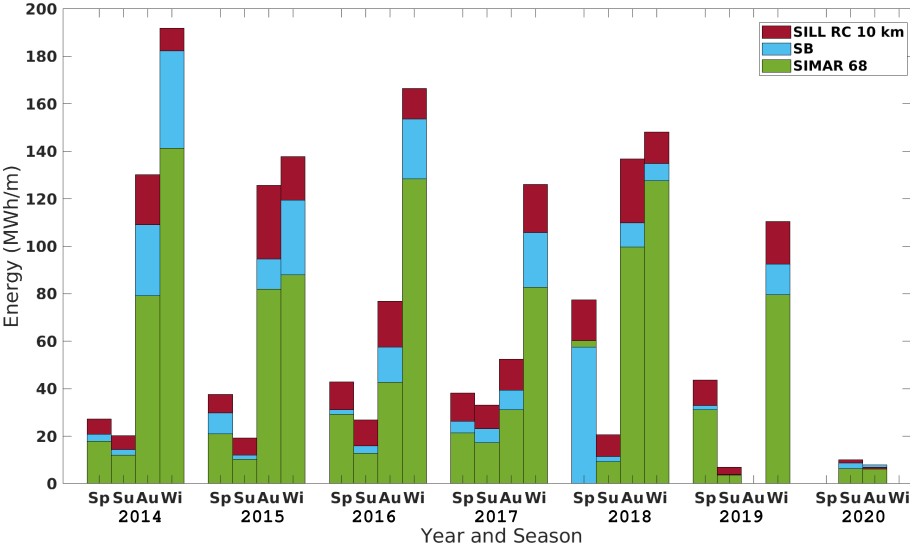

**Figure 7.** Superimposed bars of total wave energy estimated with SILL (**red**); S68 (**green**); and SB (**blue**) by season of the year.

### 3.2. Wave Energy Resource

Figures 8 and 9 show the energy matrices describing the wave energy resource in terms of $T_e$ and $H_{m0}$ available at the Vilán and Silleiro sites, respectively. Matrices were calculated with the paired raw data from the radars, buoys and SIMAR points (Table 3). To compare the matrices, the percentage of occurrences were calculated with the same ideal number of samples for each area (Table 1).

The most relevant differences are related to the large radars' data loss (the smallest waves), such as low percentages of occurrence and higher values of percentile power lines (Figures 8b and 9b). In addition, the limitation of the radars to estimate waves with periods between 5 and 17 s (see Section 2) determines that $T_e$ is limited to the range of 3.43–14.57 s.

In addition, the matrices of the Vilán area (Figures 8a–c) revealed that while the $H_{m0}$ values of VB increase progressively until their maximum value at $T_e = 17$ s, for the VILA matrix (b), the maximum of $H_{m0}$ is obtained for $T_e = 11$s and a slightly higher percentage of occurrence for $H_{m0}$ values between 5 and 8 m. On the other hand, S28 matrix (a) shows relevant differences compared to VILA and VB matrices, such as the extension of the $T_e$

distribution 3–19 s and the significant bias of the percentage of occurrence towards larger periods but with a smaller $H_{m0}$.

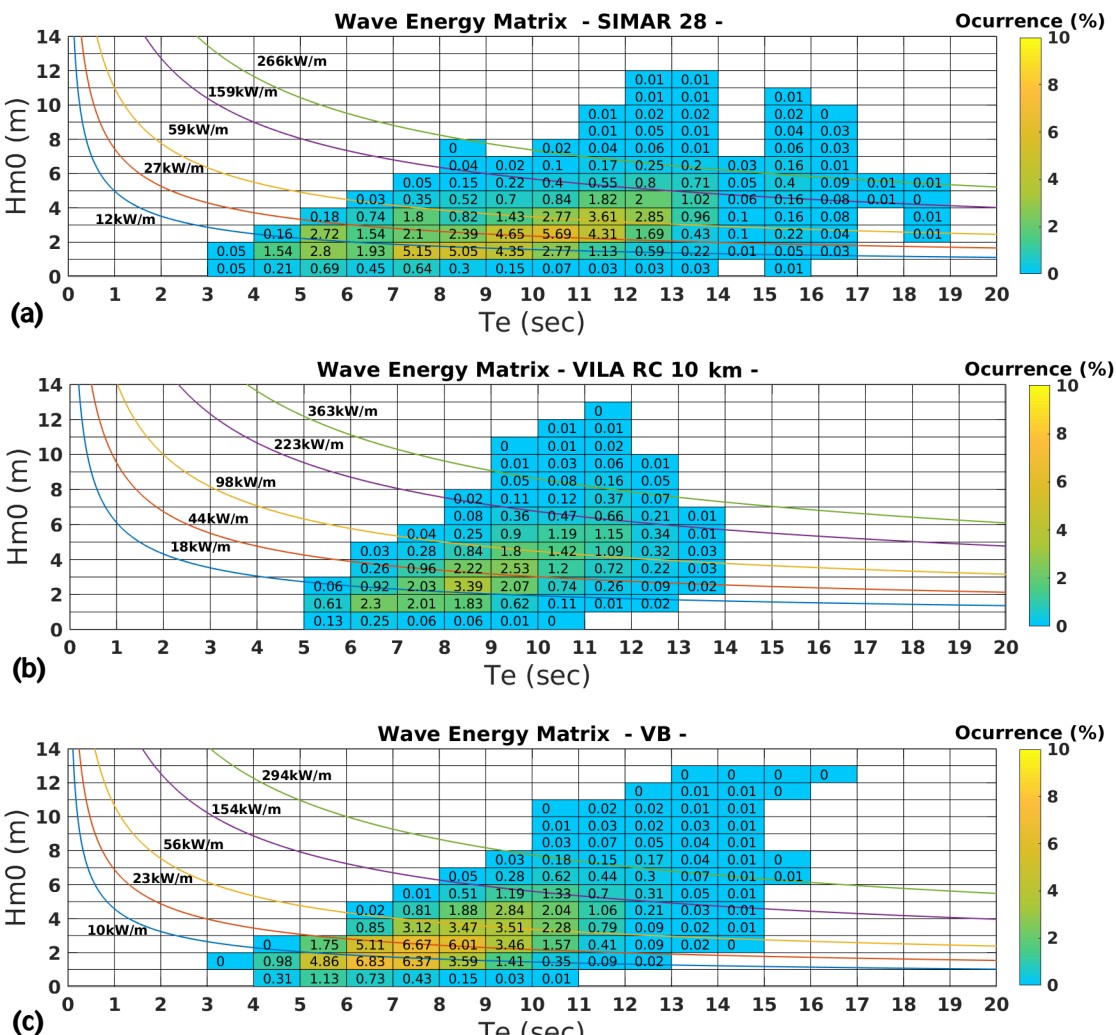

**Figure 8.** Wave energy matrices estimated with S28 (**a**); VILA RC 10 km (**b**); and VB (**c**). Numbers and colors correspond to percentage of occurrence (%), and lines are wave power (kW/m) for percentiles 25, 50, 75, 90 and 99%.

The analysis of the Silleiro area (Figure 9a–c) shows that the SILL matrix has higher percentages of occurrence between 11 and 13 s and for $H_{m0} > 8$m compared to the SB matrix. The S68 matrix distribution of the percentages of occurrence also shifted towards higher values of $T_e$, but they are lower for $H_{m0} > 4$ m and do not reach the maximum $H_{m0}$ of SB and SILL (12 m). This is clearly reflected in the lower values of S68 percentile power lines.

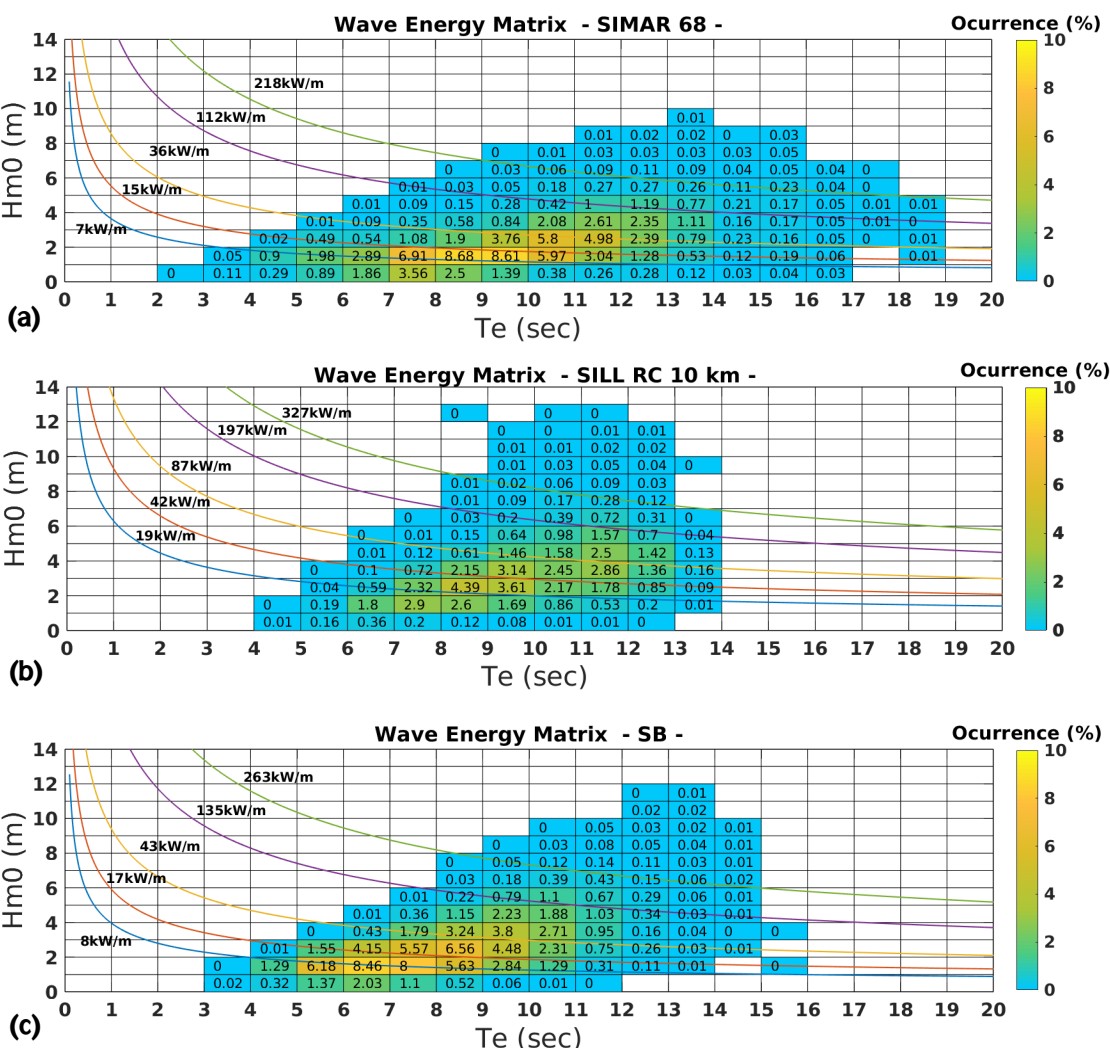

**Figure 9.** Wave energy matrices estimated with S68 (**a**); SILL RC 10 km (**b**); and SB (**c**). Numbers and colors correspond to percentage of occurrence (%), and lines are wave power (kW/m) for percentiles 25, 50, 75, 90 and 99%.

As it might be expected, given the small number of samples of VILA compared to the VB data set (13.770 vs. 29.537, Table 3), the mean wave energy (*Em*) calculated with VILA data is the smallest, both for the annual (−26%) and the seasonal periods, especially in summer (≈−50%). On the contrary, the *Em* estimated with S28 data is the largest, especially in spring and summer (19 and 14%, respectively) (Table 6a). In contrast, even with lesser samples than SB (27, 856 vs. 45, 768 samples Table 3), the *Em* obtained with SILL data is the largest, especially in autumn (15.2%). This reveals the importance of the overestimation detected in the previous validation (Figure 6). The *Em* calculated with S68 data is smaller than the one estimated with SB data, especially for the most high-energy periods (autumn and winter, −18.10% and −17.39%, respectively) (Table 6b)).

The analysis of the wave power direction was also carried out with the paired raw data sets (Table 3). Previous works for the same areas [36,39] show that main wave directions are West–Northwest (WNW) for the Vilán and Silleiro capes. The analysis of wave power by seasons revealed that only in summer in the Vilán area is there a drastic variation in the dominant direction, which VB describes as from NW to almost East (E) (Figures 10a,c). In contrast, the VILA wave rose (Figure 10b) displays the predominant directions as W and NW. Finally, S28 describes similar directions as VB. Note that most of the buoys and SIMAR data correspond to radar nulls, since it is in summer that the percentage of nulls is larger.

**Table 6.** Mean wave energy per year and season for Vilán (**a**) and Silleiro (**b**) sites.

| (a) | VILÁN SITE | | | (b) | SILLEIRO SITE | | |
|---|---|---|---|---|---|---|---|
| | Mean Wave Energy (MWh/m) | | | | Mean Wave Energy (MWh/m) | | |
| Data Sets | S28 | VILA RC 10 km | VB | Data Sets | S68 | SILL RC 10 km | SB |
| Annual | 343.28 | 240.80 | 325.75 | Annual | 246.52 | 316.83 | 292.75 |
| Spring | 45.54 | 29.24 | 38.39 | Spring | 37.94 | 44.70 | 41.97 |
| Summer | 25.93 | 11.42 | 22.70 | Summer | 20.28 | 24.23 | 22.39 |
| Autumn | 145.47 | 110.81 | 138.27 | Autumn | 73.72 | 103.69 | 90.01 |
| Winter | 132.58 | 95.42 | 131.44 | Winter | 117.18 | 148.84 | 141.84 |

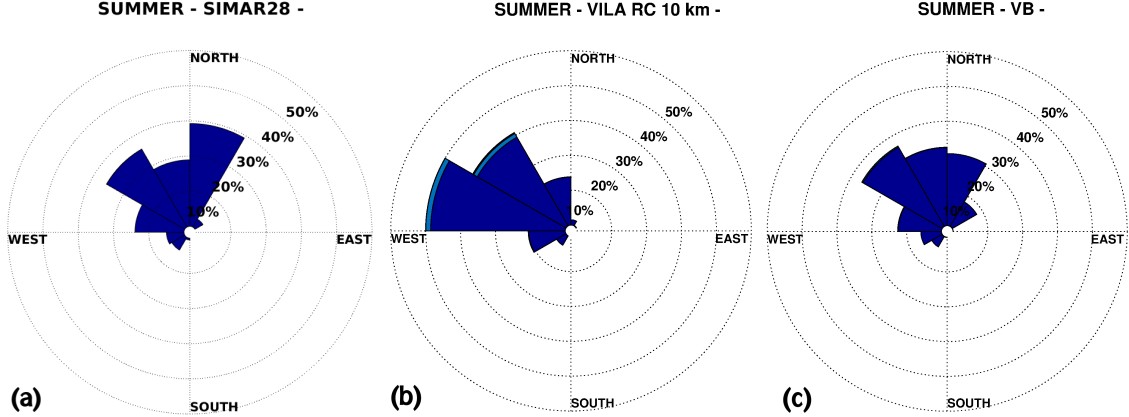

**Figure 10.** Summer wave power roses for S28 (**a**); VILA RC 10 km (**b**); and VB (**c**).

In the Silleiro area, although the predominance of the NW swell is also maintained during summer, both the buoy and the radar describe an increase in the percentage of North wave directions (330°–0°), while the SIMAR point does not show significant values with that direction (Figure 11).

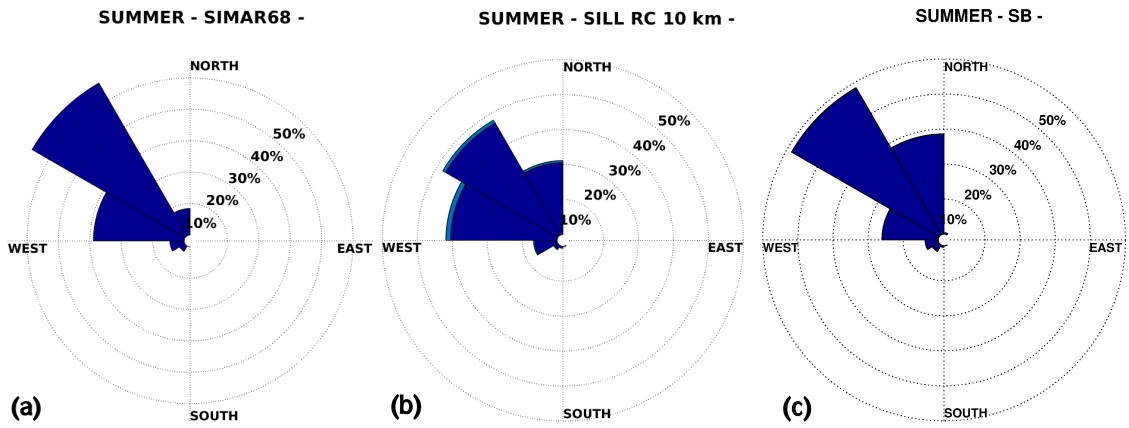

**Figure 11.** Summer wave power roses for S68 (**a**); SILL RC 10 km (**b**); and SB (**c**).

The spatial distribution of *Em* in the areas of study was described with data from the radars' 5 RCs and the WW3 model, as shown in Figures 12 and 13. In both areas, *Em* values obtained with the WW3 are larger than for the rest of the data sources except for the SILL RCs, which show similar values, except in summer when the radar data loss is especially noticeable (Figure 13h). As expected, the WW3 data show a positive gradient of energy with increasing distance from the coast and perpendicular to it. The latter highlights the handicap of the radars since measurements are obtained along arcs spanning areas with a

different distance to the shore and hence with different bathymetry and positions relative
to the coastline. Despite this, note that along the bisector of the RCs, the positive gradient
of *Em*, with some exceptions as in the case of VILA, the last RC in autumn (Figure 12i) and
the last two for the annual data (Figure 12f) and in the case of SILL, in spring (Figure 13g).
Likely, this energy loss of the farthest RCs is due to the larger percentage of nulls samples
of their data series [39].

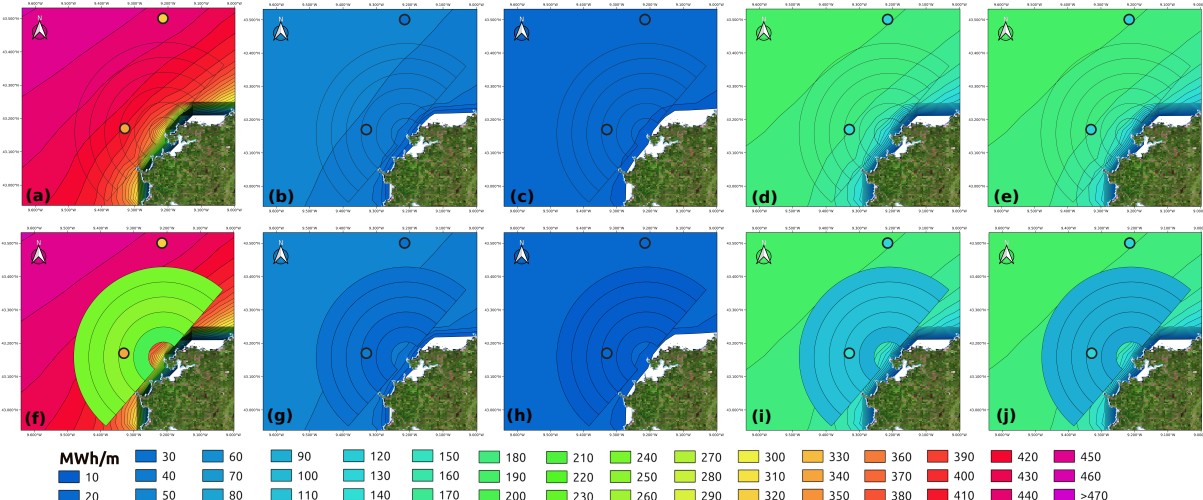

**Figure 12.** Mean annual and seasonal wave energy for the Vilán site. From left to right, the annual (**a**); spring (**b**); summer
(**c**); autumn (**d**); and winter (**e**) seasonal wave energy is shown for the WW3 model. S28 and VB values are shown within
small circles, and radar range cells values are only shown for comparison in the lower row (**f**–**j**).

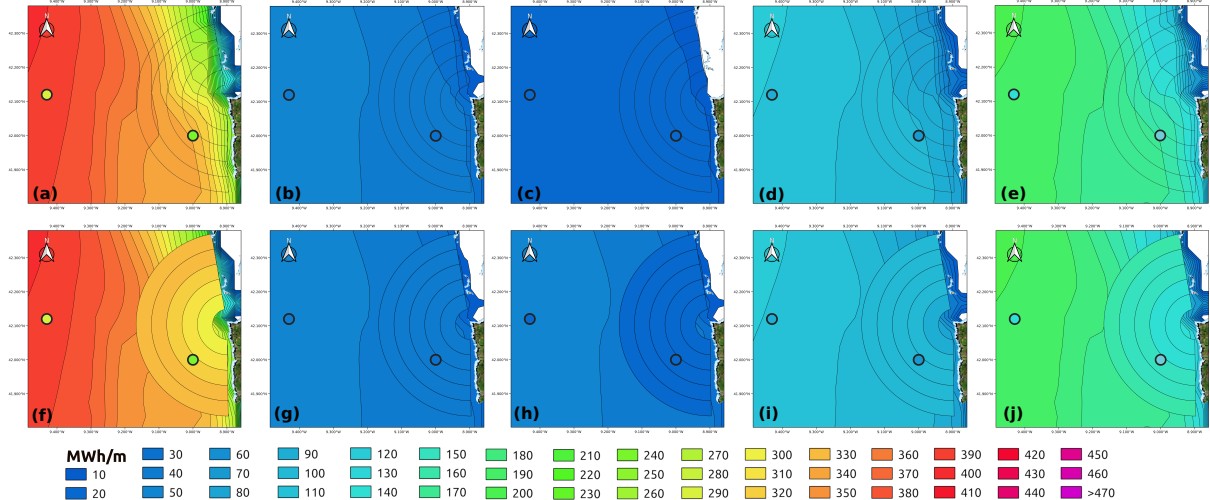

**Figure 13.** Mean annual and seasonal wave energy for the Silleiro site. From left to right, the annual (**a**); spring (**b**); summer
(**c**); autumn (**d**); and winter (**e**) seasonal wave energy is shown for the WW3 model. S68 and SB values are shown within
small circles, and radar range cells values are only shown for comparison in the lower row (**f**–**j**).

The wave power and mean energy spatial variability for the radar and WW3 model
data is shown in Table 7 in terms of a 'coefficient of variation' *CoV* [45]:

$$CoV = \left( \sum_{i=1}^{N} sd_i / mean_i \right) / N \tag{4}$$

where *N* is the data set size, *sd* and *mean* are the standard deviation and the mean of
simultaneous samples at different locations, respectively. For the radar case, these locations

are the RCs, and for the WW3 model, these are the grid cells which coincide with the RCs' bisector of the radar.

What first stands out is that the radars describe a significant difference between the mean energy and power $CoV$, which reveals the large dispersion in the radar power samples that are smooth when averaged over time. The highest $CoV$ value of $Em$ described by all the data sets occurs in summer and spring. Such $CoV$ value means variations around 1 MWh/m, that cannot be observed in Figures 12c and 13c due to their color scale. The $Em$ variability between the VILA RCs data (Table 7a) exceeds that described by WW3 by approximately +70% and in summer by ≈+180%. On the contrary, the variability of $Em$ described between the SILL RCs is lower than that described between the WW3 cells (≈-14%, Table 7b) except in spring and summer (≈+16% and +27%, respectively). Sea state along the SILL RCs might be more uniform because the straightness of the Silleiro coast and the lower data loss with the distance.

**Table 7.** Coefficient of variation ($CoV$) of the mean wave energy and wave power for Vilán (**a**) and Silleiro (**b**) sites.

| (a) | VILÁN SITE | | | | (b) | SILLEIRO SITE | | | |
|---|---|---|---|---|---|---|---|---|---|
| | Coefficient of Variation ($CoV$) (%) | | | | | Coefficient of Variation ($CoV$) (%) | | | |
| | Em | | Power | | | Em | | Power | |
| Data Sets | VILA RCs | WW3 Cells Bisector | VILA RCs | WW3 Cells Bisector | Data Sets | SILL RCs | WW3 Cells Bisector | SILL RCs | WW3 Cells Bisector |
| Annual/ Total | 7.12 | 4.04 | 22.97 | 5.30 | Annual/ Total | 4.34 | 5.02 | 23.36 | 6.81 |
| Spring | 9.66 | 3.78 | 24.66 | 5.28 | Spring | 5.20 | 4.46 | 25.21 | 6.69 |
| Summer | 12.11 | 4.32 | 19.30 | 6.16 | Summer | 7.67 | 6.03 | 26.56 | 8.08 |
| Autumn | 7.40 | 3.92 | 22.21 | 4.80 | Autumn | 3.94 | 4.96 | 18.58 | 6.45 |
| Winter | 5.88 | 4.16 | 21.82 | 5.03 | Winter | 4.20 | 5.06 | 20.43 | 5.93 |

### 3.3. WEC Electricity Energy Production

The annual and seasonal mean electricity production (MWh) of two WECs (Pelamis and Aquabuoy) were estimated using energy matrices calculated with radar, buoys and SIMAR model paired raw data (Figure 14). Both Pelamis and Aquabuoy power matrices describe the ranges of $H_{m0}$ as much more restrictive than the energy matrices used to describe the wave energy resource (Figures 8 and 9). Due to this, the electricity production using the three data sources was quite different to the previously calculated $Em$ (Table 6).

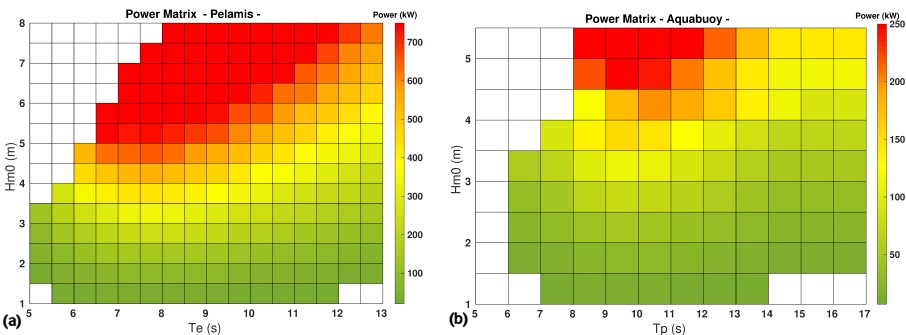

**Figure 14.** Power matrix of the wave energy converter (WEC) (based on [13]): electrical power production (kW) for the waves' parameters ($T_e$ (or $T_p$) and $H_{m0}$): (**a**) Pelamis: electric power capacity for each combination of $T_e$ and $H_{m0}$ (0.5 s × 0.5 m resolution); (**b**) Aquabuoy: electric power capacity for each combination of $T_p$ and $H_{m0}$ (1 s × 0.5 m resolution).

Hence, the electricity production calculated with VB was the largest and that calculated with VILA is the lowest (Table 8). The energy matrix of VILA describes a high percentage of occurrence at the cells between 10 and 12 s and between 3 and 6 m, which, however, are not the cells with the highest production at the Pelamis matrix power (red cells, Figure 14a). Furthermore, Aquabuoy does not produce energy with waves higher than 5.5 m (Figure 14b), so the low amount of small waves samples of VILA has a significant impact in the electricity production, especially in summer.

Likewise, the electricity production calculated with SB data is the largest and the large concentration of SILL samples at cells between 10 and 13 s and between 3 and 6 m are not used to produce too much electricity by Pelamis and Aquaboy (Table 8b,d).

**Table 8.** Mean annual and seasonal electricity production of Pelamis and Aquabuoy calculated with the radar, SIMAR, and buoy for the Vilán (**a**,**c**) and Silleiro (**b**,**d**) sites.

| | **VILAN SITE** | | | | **SILLEIRO SITE** | | |
|---|---|---|---|---|---|---|---|
| **(a)** | **Pelamis Mean Electricity Production (MWh)** | | | **(b)** | **Pelamis Mean Electricity Production (MWh)** | | |
| **Data Sets** | **S28** | **VILA RC 10 km** | **VB** | **Data Sets** | **S68** | **SILL RC 10 km** | **SB** |
| **Annual** | 1107.59 | 996.00 | 1612.53 | **Annual** | 773.39 | 1198.51 | 1421.46 |
| **Spring** | 200.00 | 146.86 | 250.16 | **Spring** | 149.98 | 199.30 | 258.66 |
| **Summer** | 135.43 | 69.94 | 158.12 | **Summer** | 99.95 | 128.74 | 145.63 |
| **Autumn** | 431.36 | 443.70 | 654.90 | **Autumn** | 223.79 | 394.71 | 442.19 |
| **Winter** | 362.94 | 362.62 | 579.31 | **Winter** | 305.86 | 492.73 | 591.41 |
| **(c)** | **Aquabuoy Mean Electricity Production (MWh)** | | | **(d)** | **Aquabuoy Mean Electricity Production (MWh)** | | |
| **Data Sets** | **S28** | **VILA RC 10 km** | **VB** | **Data Sets** | **S68** | **SILL RC 10 km** | **SB** |
| **Annual** | 362.88 | 267.22 | 442.93 | **Annual** | 260.83 | 350.55 | 415.74 |
| **Spring** | 63.90 | 43.84 | 70.39 | **Spring** | 48.86 | 62.29 | 76.81 |
| **Summer** | 41.24 | 22.03 | 45.36 | **Summer** | 30.94 | 42.28 | 44.79 |
| **Autumn** | 142.84 | 115.10 | 183.52 | **Autumn** | 79.71 | 120.13 | 134.78 |
| **Winter** | 121.95 | 93.21 | 153.82 | **Winter** | 104.05 | 131.34 | 164.88 |

*3.4. Filling Out Buoy Data Series*

To analyze the impact of the gaps in the VB data series and to use data from VILA to complete the former, new values of mean wave energy were calculated. This analysis was not carried out for SB as it does not have as many gaps as VB and because of the high overestimation of the wave energy described for SILL. All data from VB and S28 were used (Table 1), and a new data series was built combining VB and VILA samples.

Note that while the S28 data series has almost the same size (35,666) as the ideal one (35,879), VB has much less samples (33,637) (Table 1), due to the data gaps during two springs (April 2015 and April 2019) and two winters (January–February 2014 and February–March 2019), Figure 2).

Thus, the differences between the data of VB and S28 are larger (Table 9), approximately −16% for the annual *Em* set and ≈21% in winter. However, when the VB data series is filled out with VILA samples, the relative differences with S28 estimates become like those computed with paired data (Table 6).

**Table 9.** Mean annual and seasonal wave energy calculated for S28, VB and a data set resulting of filling VB data series with VILA samples (VB + VILA).

| VILÁN SITE | | | |
|---|---|---|---|
| Mean Wave Energy (MWh/m) | | | |
| Data Sets | S28 | VB | VB + VILA |
| Annual | 417.28 | 350.75 | 404.00 |
| Spring | 54.63 | 44.05 | 47.05 |
| Summer | 31.78 | 28.18 | 28.29 |
| Autumn | 145.99 | 139.43 | 139.52 |
| Winter | 180.87 | 142.02 | 183.37 |

## 4. Discussion

Part of both radars' overestimation of $H_{m0}$ and $T_e$ (and so wave power) can be considered linked to the limitations inherent to these radars' work frequency (4.5 MHz) to detect small waves [44,50,51], due to the low-energy sea states not generating second-order peaks in the Doppler spectrum. Thus, for periods of low-energy sea states (either for small periods or height), when multimodality is usually more apparent [17], the spectrum estimated by the radars is biased towards the higher-energy part of the wave spectrum [39]. Furthermore, for a calm sea, the result will be a null sample due to a lack of sufficient signal to perform any wave parameter calculation. However, radars calculate more accurately the highest-energy waves since the second-order peaks of the Doppler spectrum emerge clearly above the signal noise, avoiding saturation until waves reach approximately 20 m wave height [50].

The slight wave power overestimation described by VILA (Table 4) in the statistical validation is due to the overestimation of $H_{m0}$ and $T_e$. This was also shown by the larger percentage of occurrences of waves with $T_e$ between 9 and 12 s (Figure 8). However, when the mean wave energy was calculated with paired raw data, VILA parameters were lower than VB and S28 ones, due to the high number of nulls, especially in summer when these are most abundant (Table 6a). This also has a relevant impact on the summer predominant wave direction, (VILA data displayed as WNW and VB data as W to NEE, Figure 10). However, regarding these wave direction differences, other factors should be added, such as the radar operation limits (coastal and wave bearing) and the location, which restrict its capacity to detect the NNE waves [39]. Likewise, S28 samples do not describe directions further East than 30°, which could be explained by sheltering from coastal terrain [74].

The high spatial variability between VILA RCs (Table 7a) may be due to the radar data loss (nulls and fails), since they increase with the distance to the radar [39]. However, other contributions for such spatial variability could be both the heterogeneity of the wave regime around Cape Vilán, which spans from SW to NE directions [6,36,39], and the roughness of the shore, which could induce some irregularity in the sea states within the RCs, especially for the farthest. Hence, the resulting wave parameters of each RC can be altered [39,56] and increase the variability.

The total and mean energy (Figure 4, Table 6a) calculated with S28 are slightly higher than the VB estimates. This seems to be due to the relevant overestimation of $T_e$ but on the other hand, this is compensated by a slight underestimation of $H_{m0}$, (Figure 8a).

The use of VILA samples to fill the gaps in the VB data series (Table 9) can be very useful due to buoy maintenance (high waves), as it is easier to obtain reliable radar data. However, the differences detected in the description of the wave energy resource could influence the calculation of the electricity produced by the WECs.

The large SILL overestimation of wave power and energy described during the statistical validation (Table 5) was due to the overestimation of $H_{m0}$ and $T_e$ (Figure 3c), which is not fully compensated by the data loss, since the percentage of SILL nulls is very low for $H_{m0} > 3$ m. In addition, the SILL energy matrix described a larger abundance of

high-energy combinations of $T_e$ and $H_{m0}$ (Figure 9). As a consequence, the mean energy estimated with SILL data was still larger than that estimated with SB data, especially in autumn (Table 6b).

The agreement of SILL values with the spatial variability calculated with the WW3 model (Table 7b), compared to VILA data, could be related to the straightness of the Silleiro shoreline and the predominance of homogeneous sea states, which favour the uniformity observed for all the RCs.

S68 data describe a large dispersion of periods but also a significant underestimation of the largest $H_{m0}$ values compared to SB (Figure 9), which results in lower mean wave energy in autumn and winter (Table 6b). This is related to the SIMAR model trend to underestimate the most extreme wave heights [70].

Despite the differences between the wave data of both study areas, the discussion of the results of the electricity produced by the WECs is quite similar (Table 8). The WECs' production rates described in terms of their power matrices (Figure 14) limit the exploitation of the wave resource described by each data source [12–14].

Therefore, in the case of the radars, the swell responsible for the overestimation of the mean wave energy, especially in the case of SILL, does not correspond to the highest production rates of the WECs. Data loss due to nulls, especially in the case of VILA, is very meaningful for the electricity production of WECs such as Aquabuoy, which has a low $H_{m0}$ limit (5.5 m). In the case of the SIMAR points, the number of samples that do not contribute to the WECs electricity production is even larger (Figures 8, 9 and 14).

## 5. Conclusions

High-frequency radars, although not in situ, are considered as direct wave measurement devices, with high resolution and coverage according to the characteristics of each model. The objective of this work was to evaluate their use for wave energy resource assessment. The results shown here must be considered as a preliminary study due to the discontinuous time series.

Wave data from two CODAR Seasonde model radars were used to evaluate the wave resource offshore Vilán and Silleiro capes (NW Galician coast, Spain). In summary, the wave resource was described in terms of the annual and seasonal wave power and mean energy. The electricity production of two wave energy converters was shown. To validate the results, the wave resource was also described using wave data from two buoys, namely the two SIMAR points and the WW3 model.

The limitation of these radars to estimate small waves [39,51] seems to condition to some extent the resource estimation, especially in summer. In contrast, these radars offer greater reliability during periods of high-energy swell and may be useful to fill out the buoys data series when they are not operational.

HF radars and the SIMAR model have shown significant bias to larger values of $T_e$ that influence the estimation of the resource, and to a greater extent, the estimation of the electrical energy produced by the WECs.

Regarding this, we consider that a partitioned analysis of the wave spectra might be necessary. On the one hand, this would allow to elaborate a more complete description of the wave period, which could be useful to use and validate the data from buoys and HF radars [16,39]. On the other hand, this method might help to make accurate estimations of the WECs' production, since they are highly dependent on wave characteristics [10,11,17]. For the Seasonde radars model, it is expected that a new software will allow to estimate the multimodality of the wave spectrum [52] and thus this sort of analysis.

Finally, the radar developers recommend the design of a switch frequency radar to diminish data loss, which could measure both the smallest and the largest waves [50].

**Author Contributions:** Conceptualization, A.B.; methodology, A.B.; software, A.B.; research, A.B.; data analysis, A.B.; writing original draft preparation, A.B.; writing, review and editing, A.B. and V.P.-M.; supervision, V.P.-M. All authors have read and agreed to the published version of the manuscript.

**Funding:** This research was funded by INTERREG V-A Spain–Portugal (POCTEP) project *RADAR ON RAIA* (0461-RADAR ON RAIA-1-E) co-funded by the European Regional Development Fund (ERDF) (EU).

**Acknowledgments:** The authors gratefully acknowledge INTECMAR, Consellería do Mar, Xunta de Galicia (http://www.intecmar.gal, accessed on 20 May 2021) through RAIA Coastal Observatory for providing the Vilán HF radar data, Puertos del Estado (http://www.puertos.es, accessed on 20 May 2021) for providing Silleiro HF radar data and the SIMAR points and buoys data (REDEXT) and MeteoGalicia (https://www.meteogalicia.gal, accessed on 20 May 2021) for providing WW3 data.

**Conflicts of Interest:** The authors declare no conflict of interest.

## Abbreviations

The following abbreviations are used in this manuscript:

| | |
|---|---|
| $CoV$ | Coefficient of variation |
| $D_m$ | Mean wave direction |
| $Em$ | Mean wave energy |
| $H_{m0}$ | Spectral significant wave height |
| $T_c$ | Radar centroid period |
| $T_e$ | Energy period |
| $T_p$ | Peak period |
| AEMET | Agencia Estatal de Meteorología (Governmental Meteorological Agency in Spanish) |
| CL | Coastline limits |
| GFS | Global Forecast System |
| HF | High frequency |
| HIRLAM | High Resolution Local Area Modeling |
| P | Wave power |
| PdE | Puertos del Estado (Ports of the State in Spanish) |
| R | Lineal correlation index |
| RC | Radar range cell |
| RMSE | Root mean square error |
| S28 | SIMAR point 3004028 |
| S68 | SIMAR point 1044068 |
| SB | Silleiro buoy |
| SILL | Radar of Silleiro cape |
| SIMAR | Simulación Marina (Marine Simulation in Spanish) |
| SWAN | Simulating waves nearshore |
| VB | Vilano-Sisargas buoy |
| VILA | Radar of Vilán cape |
| WAM | Wave modeling |
| WANA | Waves analysis |
| WB | Wave bearing |
| WEC | Wave energy converter |
| WERA | Wellen radar |
| WW3 | WaveWatch III model |

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
