# Peer review of "HF Radars for Wave Energy Resource Assessment Offshore NW Spain"

_remotesensing, doi:10.3390/rs13112070_

Round 1

Reviewer 1 Report

     This manuscript describes a preliminary analysis of using two HF radars wave data
 for wave energy resource assessment. The resulting wave characterization has been used to estimate the electricity production of two wave energy converters. Results are validated against the wave data from two buoys and two numerical models. The key differences between the radars and other sources of data results are detailed. The statistical validation reveals limitations of the radar results; the authors outline ways to improve the radar output, including the use of multistatic wave spectral models and the use of a switch-frequency radar to use a higher radar transmit frequency for short waves and a lower transmit frequency for long waves. The analysis methods are interesting and can be used in the future to assess the use of HF radar wave results for wave energy resources.

I recommend that the manuscript is accepted for publication in Remote Sensing. However, before the formal acceptance, the following issues in the manuscript should be clarified:

(1) There are numerous acronyms used throughout the manuscript e.g. PdE, CL 
, Hm0 
, SB. It would be helpful include a table defining the acronyms that the reader can refer to.

(2) The paper states on Page 4 that the coastal limits are applied to wave directions. As far as I know this is not the case; please supply a reference for this statement.

(3) The paper doesn’t appear to have precise definitions for wave power and wave energy (e.g. lines 26 and 27). Since these quantities are not in SI (metric) units they need to be more clearly defined. The text needs to be reviewed to ensure consistency of the usage of the terms and of the units. If common usage mixes the terms, that should be mentioned to avoid confusing the general reader.

Author Response

(1) There are numerous acronyms used throughout the manuscript e.g. PdE, CL 
, Hm0 
, SB. It would be helpful include a table defining the acronyms that the reader can refer to.

Response 1: Thanks for the suggestion, we have added a new abbreviation section.

(2) The paper states on Page 4 that the coastal limits are applied to wave directions. As far as I know this is not the case; please supply a reference for this statement.

Response 2: The Seasonde software requires some wave bearing limits in order to help the model fitting. This can be found in CODAR Configuration files. Also, Lipa & Nyden 2005 describe this issue in the Assumptions section.

https://doi.org/10.1109/JOE.2004.839929

http://support.codar.com/Technicians_Information_Page_for_SeaSondes/Configuring_Site_files/WaveModelSetDirectionLimits.pdf

(3) The paper doesn’t appear to have precise definitions for wave power and wave energy (e.g. lines 26 and 27). Since these quantities are not in SI (metric) units they need to be more clearly defined. The text needs to be reviewed to ensure consistency of the usage of the terms and of the units. If common usage mixes the terms, that should be mentioned to avoid confusing the general reader.

Response 3: Thanks for your corrections, we have modified lines 26 and 27 and review the text. Hopefully, it will be more clear.

Reviewer 2 Report

General impressions:

This paper is a preliminary study of the use of Doppler analysis in CODAR HF radar systems for predicting the inputs and outputs of wave energy converters. In this study, HR-radar-derived ocean wave parameters (specifically wave power and energy per unit range, significant wave height, and mean wave period) are validated against in-situ buoy and model data. The authors’ methodology, analysis, and conclusions are sound and logical. The data and analysis presented here is certainly important for wave energy converter deployment, so some additional clean-up of the paper is merited to make sure the authors’ points are communicated clearly to the community.

Minor comments:

The paper has a number of grammatical errors throughout and could use a good proof-read before the next revision is submitted.

There are a number of acronyms which have been left undefined. Among them are WAN, SWAN, TAC, WAM, HIRLAM, and GFS.

Line 128: Could the authors comment on the angular accuracy of direction-finding algorithm for the HF radar system?

Line 140: Modern oceanographers prefer to use a more recent wave elevation spectrum model such as the Elfouhaily spectrum. Can the authors comments on the validity of using Pierson-Moskowitz over other spectral models? I should think that a wave spectrum allowing inputs of fetch, inverse wave age, and/or bathymetry would be more applicable in a coastal region given that P-M assumes a fully-developed sea with infinite fetch. Perhaps this is simply a shortcoming of the Seasonde model; the modality of the model is addressed in the conclusions.

Line 151: Could the authors elaborate further on the conditions under which a radar retrieval is considered a “failure”? This is not immediately clear to me. i.e., what determines the thresholds given by the bullet points?

Line 212: How applicable is the deep-water approximation in these coastal areas? Again, I do realize this is probably a shortcoming of the Seasonde model.

Line 212: Upon doing a unit check I think this is meant to be power per unit range. This should be stated explicitly, as it is not truly a power quantity, which has units of Watts.

Line 216: This description is unclear. Does “clean” radar data mean that the nulls are removed? If so, why does Table 3 also report “clean” samples in the raw data?

Line 235: It should be explicitly stated before the equation that Em is the mean wave energy per unit range to denote its significance.

Line 304: The dashes look like negative signs, but I don’t think this is intended.

Table 6: Units should read MWh/m (uppercase W for Watts).

Author Response

(1) The paper has a number of grammatical errors throughout and could use a good proof-read before the next revision is submitted.

Response 1: Thanks for the comment. We have made a new grammatical revision.

(2) There are a number of acronyms which have been left undefined. Among them are WAN, SWAN, TAC, WAM, HIRLAM, and GFS.

Response 2: Yes, we apologize for that. We have added the explanations and a new abbreviation section.

(3) Line 128: Could the authors comment on the angular accuracy of direction-finding algorithm for the HF radar system?

Response 3: We don't have a clear answer for that. According to CODAR this could be around 2º. However, there are other conditions that determine this accuracy, such as the antenna pattern used or the angle of the signal (Paduan et al, 2006).DOI: 10.1109/JOE.2006.886195 We have added a short sentence in the text.

(4) Line 140: Modern oceanographers prefer to use a more recent wave elevation spectrum model such as the Elfouhaily spectrum. Can the authors comments on the validity of using Pierson-Moskowitz over other spectral models? I should think that a wave spectrum allowing inputs of fetch, inverse wave age, and/or bathymetry would be more applicable in a coastal region given that P-M assumes a fully-developed sea with infinite fetch. Perhaps this is simply a shortcoming of the Seasonde model; the modality of the model is addressed in the conclusions.

Response 4: Thank you, we appreciate this note, and we agree. However, as you said the use of Pierson-Moskowitz model is a CODAR-Seasonde feature. We cannot select a different model in the software.

(5) Line 151: Could the authors elaborate further on the conditions under which a radar retrieval is considered a “failure”? This is not immediately clear to me. i.e., what determines the thresholds given by the bullet points?

Response 5: These thresholds were described in our previous work Basañez et al, 2020, so we have not detailed them here. In that work, we found out that most of the radar data inside these limits have a large error regarding spectral significant wave height and direction. We have clarified this point in the text.

(6) Line 212: How applicable is the deep-water approximation in these coastal areas? Again, I do realize this is probably a shortcoming of the Seasonde model.

Response 6: Yes, this assumption is a Seasonde requirement so far and also for the data retrieved from the SIMAR model. We believe that most of the time, and most of the area of the radars range cells studied can be considered deep waters.

(7) Line 212: Upon doing a unit check I think this is meant to be power per unit range. This should be stated explicitly, as it is not truly a power quantity, which has units of Watts.

Response 7: Thank you for noticing this, we have added the wave power units (W/m)

(8) Line 216: This description is unclear. Does “clean” radar data mean that the nulls are removed? If so, why does Table 3 also report “clean” samples in the raw data?

Response 8: We have used the data in two different ways; First, only paired clean data are used for statistical validation, as usual. For the rest of calculations, we have paired all the data available (the raw data), but only part of the resulting paired data sets is valid, so these are the clean samples of Table 3. We have modified the text in the manuscript, we hope it is clearer now.

(9) Line 235: It should be explicitly stated before the equation that Em is the mean wave energy per unit range to denote its significance.

Response 9: Thanks for the suggestion. We have added an explanatory line.

(10) Line 304: The dashes look like negative signs, but I don’t think this is intended.

Response 10: It is intended, for example, the negative sign -26% means 26% less energy calculated with the radar compared to the buoy.

(11) Table 6: Units should read MWh/m (uppercase W for Watts).

Response 11: Thanks, we have changed it.

Round 2

Reviewer 1 Report

Thanks for the interesting and useful paper!

Author Response

Thank you!

Ana